# Performance Analysis of On-Demand Scheduling with and without Network Coding in Wireless Broadcast

**G. G. Md. Nawaz Ali** [1,*] , **Victor C.S. Lee** [2], **Yuxuan Meng** [2], **Peter H. J. Chong** [3] **and Jun Chen** [4]

[1]  Department of Applied Computer Science, University of Charleston, Charleston, WV 25304, USA
[2]  Department of Computer Science, City University of Hong Kong, Kowloon, Hong Kong;
    csvlee@cityu.edu.hk (V.C.S.L.); yxmengwhu@sina.com (Y.M.)
[3]  Department of Electrical and Electronic Engineering, Auckland University of Technology, Auckland 1010,
    New Zealand; peter.chong@aut.ac.nz
[4]  School of Information Management, Wuhan University, Wuhan 430072, Hubei, China;
    christina_cj@whu.edu.cn
*   Correspondence: ggmdnawazali@ucwv.edu

**Abstract:** On-demand broadcast is a scalable approach to disseminating information to a large population of clients while satisfying dynamic needs of clients, such as in vehicular networks. However, in conventional broadcast approaches, only one data item can be retrieved by clients in one broadcast tick. To further improve the efficiency of wireless bandwidth, in this work, we conduct a comprehensive study on incorporating network coding with representative on-demand scheduling algorithms while preserving their original scheduling criteria. In particular, a graph model is derived to maximize the coding benefit based on the clients' requested and cached data items. Furthermore, we propose a heuristic coding-based approach, which is applicable for all the on-demand scheduling algorithms with low computational complexity. In addition, based on various application requirements, we classify the existing on-demand scheduling algorithms into three groups—real-time, non-real-time and stretch optimal. In view of different application-specific objectives, we implement the coding versions of representative algorithms in each group. Extensive simulation results conclusively demonstrate the superiority of coding versions of algorithms against their non-coding versions on achieving their respective scheduling objectives.

**Keywords:** network coding; on-demand broadcasting; data scheduling algorithm; performance evaluation

## 1. Introduction

Data broadcast is an attractive solution for large-scale data dissemination in wireless communication environments. With ever-expanding data-centric mobile applications and the recent advances in wireless communication technologies, data broadcast techniques have attracted much attention in a variety of mobile computing environments, such as vehicular networks, mobile ad hoc networks etc. [1–3]. Generally, push-based and pull-based are two commonly used broadcast approaches [4,5]. Push-based broadcast (broadcast is done periodically based on a predefined pattern) is suitable for applications with small database sizes and stable data access patterns, such as emergency message broadcast from the backend server [6,7]. In contrast, pull-based broadcast also known as on-demand broadcast disseminates data items based on clients' request patterns, such as traffic update query in traffic navigation systems [8]. In other words, in on-demand broadcast approach, the scheduling decisions are made online, which is more suitable for dynamic and large-scale data

dissemination services [9]. In this work, we focus on further improving the bandwidth efficiency of on-demand broadcast by exploiting network coding.

Several previous studies have focused on on-demand broadcasting [9–11], which have different application-specific performance objectives. In a broadcast tick (transmission time of a unit-sized data item), these strategies only able to broadcast a single data item. This restriction hinders the further improvement of broadcast bandwidth efficiency. In contrast, network coding [12] can transmit multiple data items through an encoded packet in a broadcast tick. We observe that an inherited advantage of on-demand broadcast is having the feedback information from clients to the server, and hence applying network coding in on-demand broadcast does not require additional overhead for explicit feedback.

A variety of network coding techniques have been studied. Random linear network coding (RLNC) [13] is suitable for push-based broadcast, especially for broadcasting popular contents which may serve a large population of clients [6,14], such as periodic safety message, traffic information, etc. However, RLNC is not adaptive to customized services, such as mobile infotainment, gas station query, parking space query, etc. To support such services, the server needs to receive explicit on-demand requests from clients, so that it can provide services by encoding corresponding data items accordingly. XOR ($\oplus$)-based network coding is a suitable candidate for such cases, as it has very low encoding and decoding overhead and it is simple to be implemented [15]. Many studies have applied XOR-based coding for improving network throughput [15–17] and enhancing packet delivery ratio [18]. Chaudhry and Sprintson [19] introduced the index coding problem in wireless multicast network leveraging the network coding and opportunistic listening techniques. Wang et al. [20] studied the network coding-based approach for minimizing deadline misses of real-time multi-item requests. Chen et al. [21] proposed a decoding-oriented cache management scheme in coding-assisted broadcast for reducing the overall response time. Asghari et al. [22] proposed a coded multicast delivery strategy for reducing the bandwidth usages in content placement problem.

With respect to the different performance objectives, the on-demand scheduling algorithms can be classified into three groups: *real-time*, *non-real-time* and *stretch optimal* scheduling algorithms. Specifically, in real-time scheduling, the primary objective is to minimize the deadline miss ratio, namely serving as many requests as possible before their deadlines expire. For instance, in vehicular networks, real-time traffic information must be broadcast to vehicles with certain time-constraint (e.g., must serve before a vehicle exits the service range of a road-side unit) [23,24]. Representative real-time scheduling algorithms include EDF (Earliest Deadline First) [25] and SIN (Slack time Inverse Number of pending requests) [9]. In non-real-time scheduling, the primary objective is to minimize the average response time, namely serving the requests as soon as possible. For instance, mobile clients wish to receive non-real-time infotainment information as soon as possible [10]. Representative non-real-time scheduling algorithms include FCFS (Fist Come First Served) [26], MRF (Most Requested First) [27], LWF (Longest Wait First) [27] and R × W (Number of Pending Request Multiply Waiting Time) [4]. In stretch optimal scheduling, the primary objective is to minimize the stretch, which is the ratio of the response time to the service time, where the service time refers to the transmission time of the data items. For instance, fairness treatment to different size requests in satellite networks, wireless LANs, cellular networks etc. [28]. Representative stretch optimal scheduling algorithms include LTSF (Longest Total Stretch First) [29] and STOBS (Summary Table On-demand Broadcast Scheduler) [28].

Although existing studies have considered to improve scheduling performance with respect to different application-specific objectives, they do not fully use the advantage of on-demand broadcast, where the server can be aware of the requested and cached data items of mobile clients. The impact of network coding on different scheduling objectives highly depends on the applied coding strategy at the server. In order to evaluate possible benefits brought by network coding, apparently, it is desirable to preserve the original scheduling criteria when applying network coding to existing on-demand scheduling algorithms. In this work, we will investigate the potential benefit of applying network coding into different groups of on-demand scheduling algorithms. The major contributions of this work are stated as follows.

- We present a graph model, where a CR-graph (Cache-Request-graph) [11,19] is formed based on the clients requested and cached data items for reducing redundant broadcasting.
- We transform the scheduling problem to the problem of finding maximum clique in the derived CR-graph, which is a well-known NP-complete problem in graph theory [30,31]. Accordingly, we propose a heuristic coding-based approach, which ensures that each scheduling algorithm will transform into its corresponding coding version while preserving its original scheduling criterion.
- For the three groups of scheduling algorithms, namely real-time, non-real-time and stretch optimal, we select representative solutions in each group and present the detailed implementation of their coding versions with the proposed approach.
- To evaluate the impact of network coding to different groups of algorithms, we build the simulation model for each group and define the used performance metrics.
- We give a comprehensive performance evaluation, which demonstrates the efficiency of network coding on enhancing system performance with respect to different scheduling objectives under various circumstances.

The rest of this paper is organized as follows. Section 2 reviews the related work. Section 3 presents the system model. Section 4 proposes a heuristic coding-based approach. We build the simulation model in Section 5. An extensive performance evaluation is conducted in Section 6. Finally, we conclude this work in Section 7.

## 2. Related Works

The two main components of the proposed work in this paper are the on-demand wireless data broadcast and network coding. In the following subsections, first we review some existing representative on-demand scheduling algorithms in real-time, non-real-time and stretch optimal models. Then we review some exiting works in network coding, and state the possibilities and challenges of applying network coding in on-demand wireless data broadcast.

### 2.1. Real-Time Model

In real-time applications, the requests submitted by clients are associated with deadlines, and the primary objectives of real-time scheduling is to minimize the deadline miss ratio of requests. EDF (Earliest Deadline First) [25] is a classical scheduling algorithm in real-time systems, broadcasts data item based on the urgency of the request. To achieve the lower request deadline miss ratio and access time, Hu [32] proposed to consider request urgency, service productivity and scheduling fairness in scheduling. Motivated from EDF and MRF (Most Requested First), Xu et al. [9] proposed a real-time scheduling algorithm called SIN (Slack time Inverse Number of pending requests). For time-critical services, Chung et al. [33] proposed an algorithm called SUSC (Scheduling under Sufficient Channels), which creates a broadcast program under sufficient channels. Alternatively, PAMAD (the Progressively Approaching Minimum Average Delay) is used when the number of channels is insufficient. DTIU (Dynamic Temperature Inverse Urgency) [34] is proposed to cater real-time multi-item requests, which considers both data item popularity and request urgency. Ali et al. [3] studied the performance of different scheduling algorithm under strict deadline environment. He et al. [35] proposed UPF (Urgent and Popular data item First) for scheduling real-time requests under multi-channel environment.

### 2.2. Non-Real-Time Model

In non-real-time applications, the responsiveness of the system is measured by the time from the submission of a request to the time that this request has been served. The primary objectives of non-real-time scheduling algorithms is to minimize the response time of serving requests. Many classical algorithms have been proposed for non-real-time applications. FCFS (First Come First Served) [26] broadcasts data items in the arrival order of the corresponding requests. Wong [27] proposed two popular scheduling algorithms: MRF (Most Requested First) and LWF (Longest Wait First). MRF broadcasts the data item according to the popularity of the requested data items. LWF

broadcasts the data item with the longest total waiting time. Aksoy and Franklin [4] suggested that the straightforward implementation of LWF is exhaustive and not suitable for large systems. They proposed a new scheduling scheme called R × W (Number of Pending Requests Multiply Waiting Time), which combines the benefits of MRF and FCFS. R × W broadcast the data item with the maximum R × W value. Liu and Lee [5,10] analyzed the mean response time of different existing scheduling algorithms under multi-channel environment. Polatoglou et al. [36] studied the complexity of adaptive push-based broadcast system for reducing mean response time. Lu et al. [37] studied the combined data push and data retrieval scheduling problem in multi-channel environment.

### 2.3. Stretch Optimal Model

In stretch optimal scheduling, the algorithm takes the variable sizes of data items into consideration (i.e., the service time to different data items). A metric called *stretch* is proposed in [29]. It is the ratio of the response time to the service time of the requested data item. The response time refers to the duration from the instance when the request is submitted to the time when the request is satisfied, whereas the service time is the duration for transmitting the data item. Acharya and Muthukrishnan [29] proposed a stretch optimal scheduling algorithm called LTSF (Longest Total Stretch First). In LTSF, the data item with the largest current stretch is broadcast first. Wu and Cao [38] reduced the computation overhead to make the stretch-based scheduling more scalable in on-demand broadcast environments. STOBS (Summary Tables On-demand Broadcast Scheduler) is proposed in [28], which broadcasts the data item with the maximum $\frac{R \times W}{S}$ value. Here, $R$, $W$ and $S$, respectively, denote the number of pending requests for the data item, the waiting time of the oldest request and the data item size. Lee et al. [39] proposed PRDS (Preemptive Request Deadline Size), for minimizing the response time and stretch in scheduling size variant data item.

### 2.4. Network Coding

Random linear network coding (RLNC) is initially proposed for the transmission and compression of information in multi-source multicast networks [13]. Ye et al. [6] applied RLNC for pushing popular contents from a single sever to multiple clients. Hassanabadi and Valaee [14] proposed to apply RLNC for repetitive reliable safety message rebroadcasting in the congested wireless channel. Ploumidis et al. [40] analyzed the throughput and delay in wireless mesh networks incorporating RLNC. Qu et al. [41] also studied the tradeoff between the throughput and decoding delay in applying RLNC in wireless mesh networks, and proposed a protocol named DCNC (Delay Controlled Network Coding). However, as in RLNC, the random coefficients need to be attached with the coded message, the coding overhead can be significant in a congested network. To reduce the overhead, a number of studies have considered XOR-based network coding in on-demand broadcast. Chu et al. [42] proposed On-demand Encoding (OE) algorithm for reducing clients' access time in on-demand broadcast. However, OE can only encode two data items in an encoded packet, and hence it is not flexible, and it cannot maximize the coding benefit. Chaudhry and Sprintson [19] transformed the on-demand broadcast coding problem to the Boolean satisfiability (SAT) problem. They proposed several heuristic solutions based on graph coloring and color saving algorithms. Gao et al. [43] studied XOR coding for packet retransmission in erroneous wireless medium. Chen et al. [44] proposed an on-demand broadcast algorithm called ADC (Adaptive Demand-oriented Coding) for serving multi-item requests. In [11], Chen et al. proposed ADC-1 and ADC-2, where ADC-1 considers scheduling and network coding operations separately, and in ADC-2, these two operations are considered integrally. Zhan et al. [17] proposed a model for constructing encoding packet with several data items. The proposed approach greedily finds maximal cliques based on constructed graph model to find an approximate solution to the formulated coding problem. Ali et al. [45] proposed network coded opportunistic relaying for improving data dissemination performance.

In summary, to improve the scheduling performance by further exploring the strength of on-demand broadcast, it is desirable to adopt network coding into data broadcast. With network coding,

different data items can be encoded into one packet. Since different coding strategies have different impacts on scheduling performance, in this paper, we propose a general heuristic coding-based on-demand broadcast scheme, by which the data scheduling problem can be transformed to the maximal clique problem. With the proposed approach, existing on-demand scheduling algorithms can be migrated into their corresponding coding versions while preserving their original criteria in scheduling data items. In addition, this work validates the advantage of applying network coding in on-demand scheduling through a comprehensive simulation study.

## 3. System Model

### 3.1. System Architecture

The system architecture shown in Figure 1 represents a typical on-demand data broadcast system in wireless communication medium [4]. The system consists of one server and multiple clients. When a client needs a data item and is not found in its local cache, the client submits a request via the uplink channel to the server, and senses the downlink channel for the transmission of the requested data item. According to a used underlying scheduling algorithm, the server retrieves the data items from the local database and constructs an encoded packet for broadcasting. While a server broadcasts the encoded packet, it also broadcasts the index information of the data items that are being encoded in the coded packet. Hence, when a client receives an encoded packet, it knows which data items from its local cache need to be used in decoding. A client can decode a requested data item from an encoded packet if it has cached all the other data items in the encoded packet except the requested one. Once decoding is successful, the decoded data item is then added in its local cache. The bitwise XOR (exclusive-OR) operation is commonly adopted for encoding and decoding due to its trivial computation overhead [11,42,46]. For example, as shown in Figure 1, C3 (i.e., the Client 3) has cached $d_1$. Suppose C3 requests $d_2$ and the server broadcasts an encoded packet $d_1 \oplus d_2$. C3 can decode $d_1 \oplus d_2$ and retrieve $d_2$ by such a schedule. For the cache management in a client, the LRU (Least Recently Used) cache replacement policy is adopted. A client will generate a new request only when the previous submitted request is satisfied or misses its deadline, which resembles the closed system model [11].

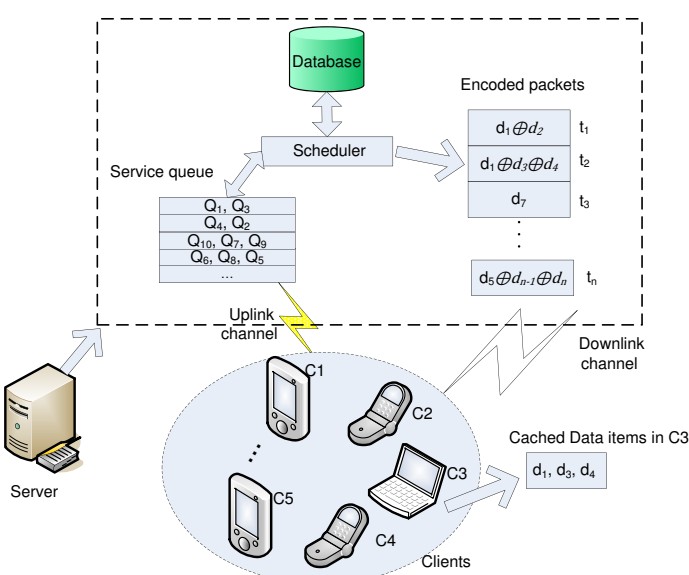

**Figure 1.** System architecture.

### 3.2. Graph Model

The key to improve the system performance for real-time, non-real-time and stretch optimal models is to disseminate a greater number of requested data items per broadcast tick. From the

network coding perspective, it needs to make the encoding decision in such a way so that more number of clients will be satisfied per coded packet and eliminates the redundant encoding. For this, assume that $G(V, E)$ is a CR-graph (Cache-Request-Graph) which is formed from the clients' cached and requested data items. The next step is to find a maximum clique which contains the maximum priority vertex based on the used scheduling algorithm. In on-demand broadcast, CR-graph construction is a practical assumption because clients can piggyback their updated cache information in the requests submitted to the server. With the knowledge of clients' cached and requested items, a graph $G(V, E)$ is constructed at the server to guide the encoding decision. A brief overview of graph construction is given below. The summary of used notations in this paper is shown in Table 1.

**Table 1.** Summary of notations.

| Notation | Description | Notes |
|---|---|---|
| $c_i$ | A client | $c_i \in C; C = \{c_1, c_2, \cdots, c_n\}$ |
| $d_j$ | A data item | $d_j \in D; D = \{d_1, d_2, \cdots, d_m\}$ |
| $R(c_i)$ | Requested data items by $c_i$ | |
| $CH(c_i)$ | Cached data items by $c_i$ | $R(c_i) \cap CH(c_i) = \varnothing$ |
| $G(V, E)$ | A graph | |
| $v_{ij}$ | A vertex representing client $c_i$ requests data item $d_j$ | $v_{ij} \in V(G), 1 \le i \le n, 1 \le j \le m$ |
| $N(d_j)$ | popularity of $d_j$ | |
| $d_{j(i)}$ | Client $c_i$ is pending for $d_j$ | |
| $t_i$ | Waiting time of $c_i$ | |
| $\sum t(d_j)$ | Summed waiting time of clients pending for $d_j$ | |
| $T_i$ | Slack time of $c_i$ | |
| $l_i$ | Size of the requested data item of $c_i$ | |
| $\delta$ | A clique | |
| $|\delta|$ | Number of vertices in $\delta$ | |
| $C_\delta$ | Set of clients covered in $\delta$ | $C_\delta = \{c_1, c_2, \cdots, c_{|\delta|}\}$ |
| $\beta(\delta)$ | Encoded packet for the clique $\delta$ | $\beta(\delta) = \{d_1 \oplus d_2 \oplus \cdots \oplus d_K\}$ |
| $K$ | Number of different data items encoded in $\beta(\delta)$ | $K \le |\delta|$ |
| $\delta_{max}$ | Maximum clique | |
| $\delta_{max}^{v_{ps}}$ | Maximal clique covering vertex $v_{ps}$ | $|\delta_{max}^{v_{ps}}| \le |\delta_{max}|$ |
| $B$ | Channel broadcast bandwidth | |

$C$ denotes the set of clients in the system, which is, $C = \{c_1, c_2, \cdots, c_n\}$, where $n$ is total number of clients. The set of data items requested by client $c_i$ is denoted by $R(c_i)$, and the set of data items cached at $c_i$ is denoted by $CH(c_i)$ $(1 \le i \le n)$. The server has a database $D$ with $m$ data items, where $d_j$ is the $j$th data item $(1 \le j \le m)$.

**Definition 1.** *Based on the clients' requested and cached data item, a graph $G(V, E)$ is constructed by the following rules:*
$V = \{v_{ij}|$ *client $c_i$ requests for item $d_j$*, $1 \le i \le n, 1 \le j \le m\}$
$E = \{(v_{i_1 j_1}, v_{i_2 j_2})| j_1 = j_2;$ *or $j_1 \ne j_2$, $d_{j_2} \in CH(c_{i_1})$, $d_{j_1} \in CH(c_{i_2})\}$*

Accordingly, the rules for constructing edges are specified below:

- $e(v_{i_1 j_1}, v_{i_2 j_2})$ with $j_1 = j_2$ means that if clients $c_{i_1}$ and $c_{i_2}$ request the same data item, there will be an edge between the two vertices $v_{i_1 j_1}$ and $v_{i_2 j_2}$.
- $e(v_{i_1 j_1}, v_{i_2 j_2})$ with $j_1 \ne j_2$, $d_{j_2} \in CH(c_{i_1})$, and $d_{j_1} \in CH(c_{i_2})$ means that if client $c_{i_1}$'s cache contains the data item being requested by client $c_{i_2}$ and vice versa, there will be an edge between vertices $v_{i_1 j_1}$ and $v_{i_2 j_2}$.

A clique, denoted by $\delta$, is a subset of the vertices in the CR-graph $G$, such that every two vertices are connected by an edge, i.e., $\delta \subseteq V(G)$.

**Definition 2.** *Let an arbitrary clique $\delta$ contains $|\delta|$ number of vertices, i.e., $\delta = \{v_{i_1 j_1}, v_{i_2 j_2}, \cdots, v_{i_{|\delta|} j_{|\delta|}}\}$. The corresponding clients are $C_\delta = \{c_1, c_2, \cdots, c_{|\delta|}\}$. The encoded packet for clique $\delta$ will be $\beta(\delta) = \{d_1 \oplus d_2 \oplus \cdots \oplus d_K\}$, where K different data items are requested by clients in $C_\delta$. Please note that $K \leq |\delta|$.*

By encoding the set of data items in $\delta$ for broadcasting, scheduling with coding can satisfy multiple requests for different data items in a broadcast tick. In contrast, scheduling without coding can only serve the requests for the same data item in a broadcast tick.

*3.3. Coding-Based On-Demand Broadcast*

Recalling $v_{ij}$ denotes a client $c_i$ for the requested data item $d_j$. For simplicity, we assume that the vertices in the graph are denoted by $v_1, v_2, \cdots, v_n$, where $n$ is number of vertices. Please note that this simple assumption does not refrain a vertex to also denote the corresponding requested data item. $v_i$ corresponds to the request $c_i$, where $v_i \in V(G)$ and $1 \leq i \leq n$. A vertex $v_i$ is associated with three weights, namely $t_i$, $l_i$, and $T_i$, which represent the waiting time, the item size and the slack time, respectively. A data item $d_j$ is also associated with a weight $N(d_j)$, which represents the popularity, namely the number of pending requests for $d_j$. The broadcast channel bandwidth is $B$. The relationships between the scheduling objectives and the properties of the cliques are stated as follows.

- Relationship 1: Real-time scheduling algorithms which aim to serve the request with the minimum slack time such as EDF, should select the largest clique $\delta$ which includes the vertex $v_i$ with the largest $\frac{1}{T_i}$ value. The other real-time algorithm SIN, also considers popularity in addition of slack time.
- Relationship 2: Non-real-time scheduling algorithms which aim to serve the request with the longest waiting time such as FCFS, should select the largest clique $\delta$ which includes the vertex $v_i$ with the largest $t_i$ value.
- Relationship 3: Non-real-time scheduling algorithms which aim to broadcast the data item with the highest popularity such as MRF, should select the largest clique $\delta$ which includes the maximum number of vertices.

    The working properties of other non-real-time algorithms are the combination of the properties of FCFS (waiting time) and MRF (data item popularity), such as LWF and R×W, both consider waiting time and data item popularity for scheduling.
- Relationship 4: Stretch optimal scheduling algorithms which aim to reduce the stretch such as LTSF, should select the clique $\delta$ which has the maximum current stretch, namely with the maximum summation of $\frac{t_i}{\frac{T_i}{B}}$ value. The other stretch optimal scheduling algorithm STOBS considers data item popularity in addition of request waiting time and data item size.

The relationships of all the algorithms are summarized in Table 2. From the above relationships it is understandable that to migrate the traditional scheduling algorithms into the network coding versions by exploiting the full coding benefits, we need to find the maximum clique $\delta_{max}$ in the graph $G(V, E)$. This is because the number of satisfying clients is equivalent to the number of vertices in $\delta_{max}$. However, finding the maximum clique is a well-known NP-complete problem in graph theory [30]. This is computationally expensive, because the new maximum clique needs to find out every time before making a broadcast decision. This turns out applying network coding in on-demand broadcast may be infeasible in practice. Hence in the following we propose a heuristic coding-based approach to convert the traditional scheduling algorithms into the coding version in polynomial time, while preserving the original scheduling criteria.

**Table 2.** Coding-based on-demand broadcast.

| Type | Algorithm | Remarks |
|---|---|---|
| Real-time | EDF | Find the largest clique $\delta$ with $T_{min}$ |
| | SIN | Find the largest clique $\delta$ with the maximum $\frac{|\delta|}{T_{min}}$ value |
| Non-real-time | FCFS | Find the largest clique $\delta$ with $t_{max}$ |
| | MRF | Find the largest clique $\delta$ |
| | LWF | Find the largest clique $\delta$ with the maximum summed waiting time |
| | $R \times W$ | Find the largest clique $\delta$ with the maximum $|\delta| \times t_{max}$ value |
| Stretch optimal | LTSF | Find the largest clique $\delta$ with the maximum $\frac{\sum_{v_i \in \delta} t_i}{max_{v_i \in \delta}\{\frac{l_i}{B}\}}$ value |
| | STOBS | Find the largest clique $\delta$ with the maximum $\frac{|\delta| * max_{v_i \in \delta} t_i}{max_{v_i \in \delta} l_i}$ value |

## 4. Proposed Heuristic Coding-Based On-Demand Broadcast

Recalling the objective of this work is to convert a traditional scheduling algorithm into the coding version with its original scheduling criteria along with maximizing the coding benefit by keeping the overhead minimal. Hence the key idea of the proposed heuristic coding-based approach is to broadcast the encoded packet, which (1) must contain the vertex corresponds to the scheduled client by the scheduling algorithm, and (2) satisfy as many clients as possible per broadcast by keeping the encoding complexity as minimal as possible. The proposed approach contains the following two important considerations.

- Find the vertex, which holds the maximum priority value of the scheduling algorithm. If a data item $d_s \in D$ of a client $c_p$ is selected to broadcast, the corresponding vertex is $v_{ps}$, we call as *selected vertex*.
- To maximizing the exploitation of network coding and reducing encoding complexity, need to find the maximal clique $\delta_{max}^{v_{ps}}$ (approximate maximum clique) which covers the selected vertex $v_{ps}$. Please note that $\delta_{max}^{v_{ps}}$ is the maximum among all the possible maximal cliques covering $v_{ps}$ in $G$. $\delta_{max}^{v_{ps}}$ will be used to form the encoded packet $\beta(\delta_{max}^{v_{ps}})$ for broadcasting.

**Lemma 1.** *For an arbitrary vertex $v_{ij} \in \delta_{max}^{v_{ps}}$, the corresponding client $c_i \in C_\delta$ can retrieve $d_j$ from the encode packet $\beta(\delta_{max}^{v_{ps}})$.*

**Proof.** As the proposed system is a closed system, which infers that one client can have only one pending request in the service queue. Hence, we can infer that all the vertices in a clique represents the unique clients. According to the clique definition, all the vertices in a clique are inter-connected. Again according to the edge definition of a CR-graph (Definition 1), a client only has edges with other clients if they request the same data item, or one client's cached data item is being requested by another client and vice versa. This implies that for a client $c_i \in C_\delta$, if $v_{ij} \in \delta_{max}^{v_{ps}}$, for $K \geq 1$, $c_i$ already has (in the cache) all the data items of $\beta(\delta_{max}^{v_{ps}})$, except $d_j$; otherwise all the vertices in $\delta_{max}^{v_{ps}}$ requested for the same data item $d_j$ and $d_j = d_s$. Hence $c_i$ can easily decode $d_j$ from $\beta(\delta_{max}^{v_{ps}})$ with its cached data items. □

**Lemma 2.** *Broadcasting the encoded packet $\beta(\delta_{max}^{v_{ps}})$ for the selected vertex $v_{ps}$, ensures the migration of a scheduling algorithm into the coding version with preserving its original scheduling criteria.*

**Proof.** The proposed coding-based system first selects the selected vertex $v_{ps}$, then finds the maximal clique $\delta_{max}^{v_{ps}}$, which covers $v_{ps}$. Please note that $v_{ps}$ is the highest priority vertex based on the scheduling criteria of a scheduling algorithm. For instance, in the real-time model, for EDF scheduling, $v_{ps}$ is vertex with the highest urgency; in the non-real-time model, for FCFS scheduling, $v_{ps}$ is vertex with the longest waiting time; in the stretch optimal model, for LTSF scheduling, $v_{ps}$ is vertex with the longest stretch. After selecting $v_{ps}$, the maximal clique $\delta_{max}^{v_{ps}}$ is found from the CR-graph $G$, which cover $v_{ps}$.

This means that encoded packet $\beta(\delta_{max}^{v_{ps}})$ must contain the selected data item $d_s$ of client $c_p$. Hence by broadcasting $\beta(\delta_{max}^{v_{ps}})$, all the scheduling algorithms' criteria are preserved in the coding version. $\quad\square$

*4.1. The Heuristic Algorithm*

To sum up, our heuristic coding-based on-demand broadcast consists of the following three key steps. The pseudo code is shown in Algorithm 1.

---

**Algorithm 1:** Heuristic coding-based on-demand broadcast.

---

1 **Step 1: CR-graph construction**

2 $G(V, E)$ is the already constructed CR-graph, and $v_{kr}$ is the vertex for a newly arrived client $c_k$, where $1 \le k \le n$, $1 \le r \le m$ ;

3 **for** *each $v_{ij} \in V(G)$* **do**

4 $\quad$ **if** *j=s* **then**

5 $\quad\quad$ $E(G) \leftarrow E(G) + e(v_{ij}, v_{kr})$;

6 $\quad$ **else if** *$d_j \in H(c_k)$ & $d_r \in H(c_i)$* **then**

7 $\quad\quad$ $E(G) \leftarrow E(G) + e(v_{ij}, v_{kr})$;

8 $\quad$ **else**

9 $\quad\quad$ /* There is no link between $v_{ij}$ and $v_{kr}$ $\qquad\qquad\qquad\qquad\qquad\qquad\qquad$ */

10 $V(G) \leftarrow V(G) + v_{kr}$;

11 **Step 2: Find the selected vertex $v_{ps}$**

12 Each vertex $v_{ij} \in V(G)$ is associated with $T_i, t_i, l_i$;

13 Each data item $d_j \in D$ is associated with $N(d_j)$;

14 Initialize $MAX \leftarrow 0$;

15 **for** *each $v_{ij} \in V(G)$* **do**

16 $\quad$ $P \leftarrow$ Invoke the underlying scheduling algorithm with $T_i, t_i, l_i, N(d_j)$;

17 $\quad$ **if** *$P > MAX$* **then**

18 $\quad\quad$ $MAX \leftarrow P$;

19 $\quad\quad$ $v_{ps} \leftarrow v_{ij}$;

20 **Step 3: Encode and broadcast the encoded packet $\beta(\delta_{max}^{v_{ps}})$**

21 Find the maximal clique $\delta_{max}^{v_{ps}}$ for the selected vertex $v_{ps}$;

22 Form the encoded packet $\beta(\delta_{max}^{v_{ps}}) \leftarrow \{d_1 \oplus d_2 \oplus \cdots \oplus d_k\}; k \le |\delta_{max}^{v_{ps}}|$;

23 Broadcast $\beta(\delta_{max}^{v_{ps}})$;

24 /* Update $G(V, E)$ $\qquad\qquad\qquad\qquad\qquad\qquad\qquad\qquad\qquad\qquad\qquad\qquad\qquad\qquad\qquad$ */

25 **for** *each $v_{ij} \in \delta_{max}^{v_{ps}}$* **do**

26 $\quad$ $V(G) \leftarrow V(G) - v_{ij}$;

---

**Step 1: CR-graph $G(V, E)$ construction**

Upon a request arrival from a client $c_k$ for the data item $d_r$, the corresponding vertex $v_{kr}$ will be added in the vertex set $V(G)$. According the edge definition as defined in Definition 1, all the possible edges are formed for $v_{kr}$ with the vertices in $V(G)$.

**Step 2: Finding the selected vertex $v_{ps}$**

Based on the associated weights of a vertex $v_i$, namely $t_i, T_i, l_i$, and the associated weight of the requested data item $d_j$, namely $N(d_j)$, a particular scheduling algorithm selects a vertex $v_{ps}$ with the highest scheduling priority value ($P$) among all the vertices in $V(G)$. For instance, EDF selects the vertex $v_{ps}$ with the lowest $T_p$ value, MRF selects a vertex $v_{ps}$ with the highest $N(d_s)$, LTSF selects

a vertex $v_{ps}$ with the highest $\frac{t_p}{\frac{l_s}{B}}$ value. The details of $v_{ps}$ selection criteria of each of the scheduling algorithms are shown in Section 4.2.

**Step 3: Encoding and broadcasting the encoded packet $\beta(\delta_{max}^{v_{ps}})$**

After finding the selected vertex $v_{ps}$, in this step the heuristic approach finds the maximal clique $\delta_{max}^{v_{ps}}$ which covers $v_{ps}$. For finding the maximal clique, we use the approach presented in [11], in which only the adjacent vertices of $v_{ps}$ (directly connected vertices with $v_{ps}$) are checked for finding the maximal clique. After finding the maximal clique $\delta_{max}^{v_{ps}}$, the next job is to encode the packet $\beta(\delta_{max}^{v_{ps}})$ which consists all the corresponding data items in $\delta_{max}^{v_{ps}}$. After broadcasting $\beta(\delta_{max}^{v_{ps}})$, the system updates the CR-graph $G$.

For $n$ number of vertices the Step 1 has $O(n)$ complexity, Step 2 must search the selected vertex $v_{ps}$ which incurs $O(n)$ complexity, and the Step 3 needs to find the maximal clique for a specific selected vertex $v_{ps}$ from the constructed (in Step 1) CR-graph $G$, which incurs $O(n^3)$ complexity. As both encoding and decoding operations are performed using bitwise XOR (exclusive-OR) technique, the decoding can be performed with one single operation at each client. Therefore, decoding only incurs constant complexity $O(1)$. Hence, for each client, the decoding complexity is $O(1)$. Therefore, the total complexity of the proposed heuristic coding-based broadcast approach is $O(n) + O(n) + O(n^3) + O(1) \approx O(n^3)$, which is practical to implement.

### 4.2. Scheduling Algorithms with Heuristic Coding Implementation

We select several representative on-demand scheduling algorithms from each group and present the implementation of their coding versions. In particular, we analyze how the coding versions of different algorithms can fulfil their original scheduling objectives.

#### 4.2.1. Real-Time Algorithms

- EDF [25]: EDF serves the request with the minimum slack time. In other words, it prioritizes the request urgency in scheduling. The network coding version of EDF, needs to find the largest clique $\delta$ which contains the vertex $v_i$ with the minimum slack time $T_{min}$ among all the vertices in graph $G$. In the heuristic coding implementation (denoted by EDF_N), the system first finds the selected vertex $v_{ps}$ with $T_{min}$. In Algorithm 1, this is done in Step 2. $v_{ps}$ is the vertex with the highest EDF scheduling priority, i.e., $v_{ps} = \{v_{ij}|T_i = max\{\frac{1}{T_1}, \frac{1}{T_2}, \cdots, \frac{1}{T_n}\}, 1 \leq j \leq m\}$. Then Step 3 searches for the maximal clique $\delta_{max}^{v_{ps}}$ to ensure that the encoded packet $\beta(\delta_{max}^{v_{ps}})$ consists the requested data item of the most urgent client.

- SIN [9]: SIN broadcasts the item with the minimum *SIN* value. The network coding version of SIN, needs to find the clique $\delta$ in $G$ with the maximum $\frac{|\delta|}{T_{min}}$ value, where $T_{min}$ is the minimum value among all $T_i$ in $\delta$. In the heuristic coding implementation (denoted by SIN_N), the system first finds the selected vertex $v_{ps}$ with the minimum $\frac{T_p}{N(d_s)}$ value, In Step 2 of Algorithm 1, the $v_{ps}$ selection is done by finding the vertex with the maximum $\frac{N(d_s)}{T_p}$ value, i.e., $v_{ps} = \left\{v_{ij}|\frac{N(d_j)}{T_i} = max\left\{\frac{N(d_{r(1)})}{T_1}, \frac{N(d_{r(2)})}{T_2}, \cdots, \frac{N(d_{r(n)})}{T_n}, 1 \leq r, j \leq m\right\}\right\}$. Then Step 3 searches for the maximal clique $\delta_{max}^{v_{ps}}$ to ensure that the encoded packet $\beta(\delta_{max}^{v_{ps}})$ consists the requested data item of the client with the minimum *SIN* value.

#### 4.2.2. Non-Real-Time Algorithms

- FCFS [26]: FCFS serves requests according to their arrival order. The network coding version of FCFS, needs to find the largest clique $\delta$ which contains the vertex $v_i$ with longest waiting time $t_{max}$ among all $t_i$ in $G$. In the heuristic coding implementation (denoted by FCFS_N), the system first finds the selected vertex $v_{ps}$ with $t_{max}$. In Step 2 of Algorithm 1, $v_{ps}$ is the vertex with the highest FCFS scheduling priority, i.e., $v_{ps} = \{v_{ij}|t_i = max\{t_1, t_2, \cdots, t_n\}, 1 \leq j \leq m\}$. Then

Step 3 searches for the maximal clique $\delta_{max}^{v_{ps}}$ to ensure that the encoded packet $\beta(\delta_{max}^{v_{ps}})$ consists the requested data item of the client with the longest waiting time.

- MRF [27]: MRF broadcasts the data item with the largest number of pending requests. The network coding version of MRF, needs to find the maximum clique $\delta_{max}$ in $G$, where $\delta_{max}$ is the maximum clique among all the possible cliques in $G$. In the heuristic coding implementation (denoted by MRF_N), the system first finds the selected vertex $v_{ps}$ with the maximum $N(d_s)$ value. In Step 2 of Algorithm 1, $v_{ps}$ is the vertex with the highest MRF scheduling priority, i.e., $v_{ps} = \{v_{ij}|N(d_j) = max\{N(d_1), N(d_2), \cdots, N(d_m)\}, 1 \leq i \leq n\}$. Then Step 3 searches for the maximal clique $\delta_{max}^{v_{ps}}$ to ensure that the encoded packet $\beta(\delta_{max}^{v_{ps}})$ consists the requested data item with the largest number of pending requests.

- LWF [27]: LWF broadcasts the data item with the largest total waiting time. The network coding version of LWF, needs to find the largest clique $\delta$ in $G$ with the maximum summed waiting time of the corresponding vertices. In the heuristic coding implementation (denoted by LWF_N), the system first finds the selected vertex $v_{ps}$ with the maximum sum waiting time ($\sum t_i, 1 \leq i \leq n$) value of data item $d_s$, denoted as $\sum t(d_s)$. In Step 2 of Algorithm 1, $v_{ps}$ is the vertex with the highest LWF scheduling priority, i.e., $v_{ps} = \{v_{ij} | \sum t(d_j) = max\{\sum t(d_1), \sum t(d_2), \cdots, \sum t(d_m)\}, 1 \leq i \leq n\}$. Then Step 3 searches for the maximal clique $\delta_{max}^{v_{ps}}$ to ensure that the encoded packet $\beta(\delta_{max}^{v_{ps}})$ consists the requested data item with the largest total waiting time of the pending requests.

- R $\times$ W [4]: R $\times$ W schedules the data item with the maximum R $\times$ W value, where $R$ is the number of pending requests for that data item and $W$ is the waiting time of the earliest request for that data item. The network coding version of R $\times$ W, needs to find the largest clique $\delta$ with the maximum $|\delta| \times t_{max}$ value, where $t_{max}$ is the maximum waiting time value among all $t_i$ in $\delta$. In the heuristic coding implementation (denoted by R $\times$ W_N), the system first finds the selected vertex $v_{ps}$ with the maximum $N(d_s) \times t_p$ value. In Step 2 of Algorithm 1, the $v_{ps}$ selection is done by finding the vertex with the maximum $N(d_s) \times t_p$ value, i.e., $v_{ps} = \{v_{ij}|N(d_j) \times t_i = max\{N(d_{r(1)}) \times t_1, N(d_{r(2)}) \times t_2, \cdots, N(d_{r(n)}) \times t_n, 1 \leq r, j \leq m\}\}$. Then Step 3 searches for the maximal clique $\delta_{max}^{v_{ps}}$ to ensure that the encoded packet $\beta(\delta_{max}^{v_{ps}})$ consists the requested data item with the maximum R $\times$ W value.

### 4.2.3. Stretch Optimal Algorithms

- LTSF [29]: LTSF broadcasts the data item with the largest total current stretch. The network coding version of LTSF, needs to select the largest clique $\delta$ with the maximum $\dfrac{\sum_{v_i \in \delta} t_i}{max_{v_i \in \delta}\{\frac{l_i}{B}\}}$ value, where $\sum_{v_i \in \delta} t_i$ is the total waiting time of the vertices in $\delta$ and $max_{v_i \in \delta}\{\frac{l_i}{B}\}$ is the service time of the vertices in $\delta$. Please note that the service time of the coded packet will be the service time of the largest data item in the clique. In the heuristic coding implementation (denoted by LTSF_N), the system first finds the selected vertex $v_{ps}$ with the largest total current stretch $\left( max \left\{ \dfrac{\sum t(d_s)}{\frac{l_p}{B}} \right\} \right)$ for the data item $d_s$. In Step 2 of Algorithm 1, the $v_{ps}$ selection is done by finding the vertex with the maximum $\left\{ \dfrac{\sum t(d_s)}{\frac{l_p}{B}} \right\}$ value, i.e., $v_{ps} = \left\{ v_{ij} \mid \left\{ \dfrac{\sum t(d_j)}{\frac{l_j}{B}} \right\} = max \left\{ \left\{ \dfrac{\sum t(d_1)}{\frac{l_1}{B}} \right\}, \left\{ \dfrac{\sum t(d_2)}{\frac{l_2}{B}} \right\}, \cdots, \left\{ \dfrac{\sum t(d_m)}{\frac{l_m}{B}} \right\}, 1 \leq i \leq n \right\} \right\}$. Then Step 3 searches for the maximal clique $\delta_{max}^{v_{ps}}$ to ensure that the encoded packet $\beta(\delta_{max}^{v_{ps}})$ consists the requested data item with the largest total current stretch.

- STOBS [28]: STOBS broadcasts the data item with the largest $\frac{R \times W}{S}$ value, where $R$, $W$ and $S$ denote, respectively, the number of pending requests, waiting time of the earliest pending request and the item size. The network coding version of STOBS, needs to find the largest clique $\delta$ with the maximum $\dfrac{|\delta| * max_{v_i \in \delta} t_i}{max_{v_i \in \delta} l_i}$ value, where $max_{v_i \in \delta} t_i$ and $max_{v_i \in \delta} l_i$ are respectively the maximum waiting time and the maximum item size among all the vertices in $\delta$. Please note that $max_{v_i \in \delta} l_i$

also denotes the size of the encoded packet. In the heuristic coding implementation (denoted by STOBS_N), the system first finds the selected vertex $v_{ps}$ with the maximum $\frac{N(d_s) \times t_p}{l_s}$ value. In Step 2 of Algorithm 1, the $v_{ps}$ selection is done by finding the vertex with the maximum $\frac{N(d_s) \times t_p}{l_s}$ value, i.e., $v_{ps} = \left\{ v_{ij} \mid \frac{N(d_j) \times t_i}{l_j} = max \left\{ \frac{N(d_{r(1)}) \times t_1}{l_{r(1)}}, \frac{N(d_{r(2)}) \times t_2}{l_{r(2)}}, \cdots, \frac{N(d_{r(n)}) \times t_n}{l_{r(n)}}, 1 \leq r, j \leq m \right\} \right\}$.

Then Step 3 searches for the maximal clique $\delta_{max}^{v_{ps}}$ to ensure that the encoded packet $\beta(\delta_{max}^{v_{ps}})$ consists the requested data item with the largest $\frac{R \times W}{S}$ value.

## 5. Simulation Model

### 5.1. Setup

The simulation model is built according to the system architecture described in Section 3. The model is implemented by CSIM19 [47]. Unless stated otherwise, the simulation is conducted under the simulator's default setting. The major explicit parameters are shown in Table 3.

**Table 3.** Simulation parameters.

| Parameter | Default | Range | Description |
|---|---|---|---|
| $ClientNum$ | 400 | 100~600 | Number of clients |
| $THINKTIME$ | 0.01 | — | Request generation interval |
| $m$ | 1000 | — | Size of the database |
| $SIZEMIN, SIZEMAX$ | 1, 30 | —, 10~60 | Min. and Max. data item size |
| $CacheSize$ | 60 | 30~180 | Client cache size |
| $\theta$ | 0.4 | 0.0~1.0 | Zipf distribution parameter |
| $\mu^-, \mu^+$ | 120, 200 | 80~160, 160~240 | Min. and Max. laxity |

A client only submits a new request when the previous submitted request is satisfied or misses its deadline, which exhibits a closed system model [11]. The request generation interval of each client is shaped by $THINKTIME$, which follows the Exponential distribution. The clients' data access pattern is shaped by the Zipf distribution [48], where the skewness is controlled by the parameter $\theta$. Specifically, when $\theta$ equals 0, it means the random distribution (all the data items have the equal access probability). With an increased value of $\theta$, the data access pattern becomes more skewed.

For real-time algorithms, the relative deadline of a request $q_i$ ($RD_i$) is computed by:

$$RD_i = (1 + uniform(\mu^-, \mu^+)) * T_i^{serv}$$

The deadline of $q_i$ ($Dl_i$) is computed by:

$$Dl_i = AT_i + RD_i$$

where $AT_i$ is the arrival time of $q_i$. $\mu^-$ and $\mu^+$ represent the minimum and the maximum laxity, respectively. $T_i^{serv}$ represents the service time of $q_i$. A request is feasible for serving as long as its slack time remains greater than its service time. The system routinely removes the infeasible requests from the service queue.

The channel bandwidth is 1 unit/tick, i.e., the server disseminates one unit-sized data item in each broadcast tick. For the real-time and the non-real-time scheduling, the item size is 1 unit. For the stretch optimal scheduling, the sizes of data items are generated using random distribution [49]:

$$DataItemSize[i] = SIZEMIN + \lfloor random(0.0, 1.0) \times (SIZEMAX - SIZEMIN + 1) \rfloor$$

where $SIZEMIN$ and $SIZEMAX$ are the minimum and the maximum data item size in the database, respectively, and $i = 1, 2, \cdots, m$.

*5.2. Metrics*

We define the following metrics to give an intensive performance analysis.

- Deadline Miss Ratio (DMR): It is the ratio of the number of deadline missed requests to the number of submitted requests. The primary objective of a real-time scheduling algorithm is to minimize the DMR.
- Average Response Time (ART): It is the average duration for getting the response from the server after submitting a request. ART measures the responsiveness of the system. The primary objective of a non-real-time scheduling algorithm is to minimize the ART.
- Average Stretch (AS): It is the ratio of the response time to the service time of the corresponding data item. It is a widely used metric in heterogeneous database environments. The primary objective of a stretch optimal scheduling algorithm is to minimize the AS.
- Average Encode Length (AEL): It is the average size of an encoded packet. It measures how a scheduling algorithm exploits the coding opportunity. The larger AEL mean more coding benefit.
- Average Broadcast Productivity (ABP): It is the average number of requests satisfied per broadcast. It measures the productivity of a broadcasted packet/item. A large ABP implies the efficient bandwidth use.

## 6. Performance Evaluation

In this section, we evaluate the performance of both the non-coding and the coding versions of real-time, non-real-time and stretch optimal scheduling algorithms in a variety of circumstances. The simulation was continued until 95% confidence interval was achieved.

*6.1. Simulation Results in Real-Time Models*

Figures 2 and 3 show the comparative performance graphs of the real-time algorithms for both the non-coding and the coding versions.

6.1.1. Impact of Workload

Figure 2a shows the impact of workload under different number of clients. When the number of clients increases, the system workload becomes heavier. As a result, more requests miss their deadlines and the algorithms' performances decline (Figure 2a). In the non-coding versions of the real-time algorithms, SIN outperforms EDF, because EDF only considers the request urgency. On the contrary, SIN considers both the request urgency and the item popularity, and hence SIN satisfies more requests than EDF (Figure 2b). The performance difference between EDF and SIN becomes more significant with an increase workload. The observations are consistent with the results reported in previous studies [9,50].

The network coding versions of EDF and SIN, namely EDF_N and SIN_N, outperform their respective non-coding versions notably in terms of achieving the lower deadline miss ratio (DMR). For instance, with the moderate system workload (default setting) this performance improvement is 28% and 19%, respectively for EDF_N and SIN_N. The main reason of this performance improvement is that the coding version achieves a higher average broadcast productivity (ABP) than the corresponding non-coding version (Figure 2b). This result shows that more requests can be satisfied by a single broadcast with network coding.

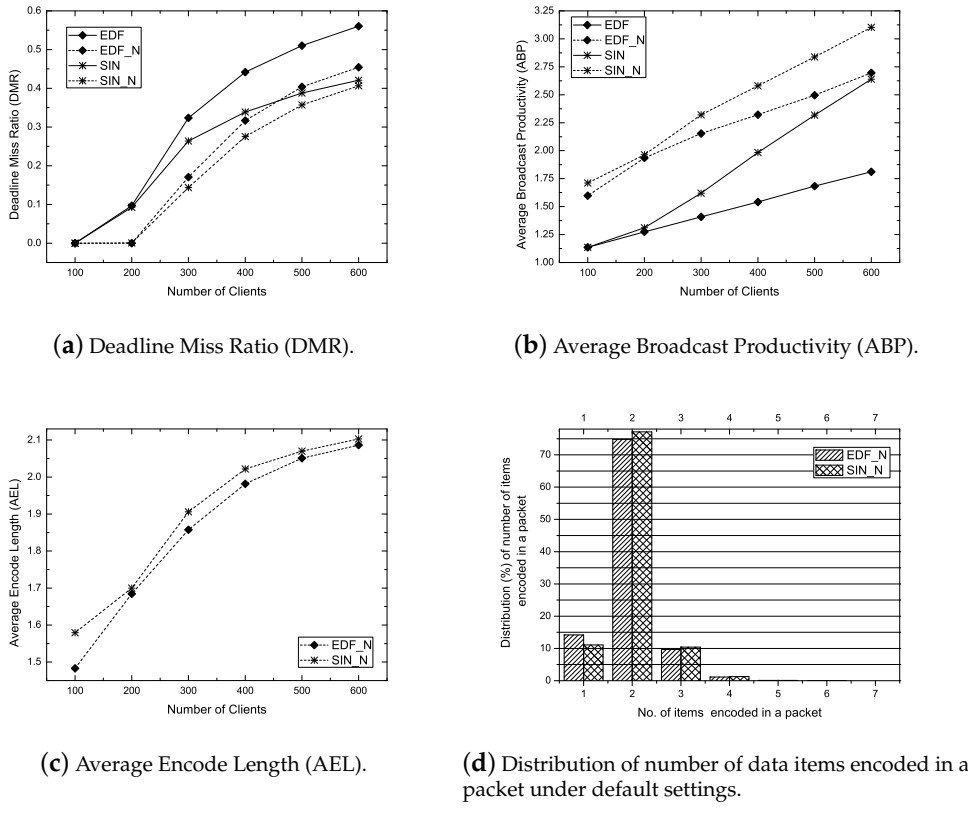

(**a**) Deadline Miss Ratio (DMR).

(**b**) Average Broadcast Productivity (ABP).

(**c**) Average Encode Length (AEL).

(**d**) Distribution of number of data items encoded in a packet under default settings.

**Figure 2.** Impact of workload on the real-time model.

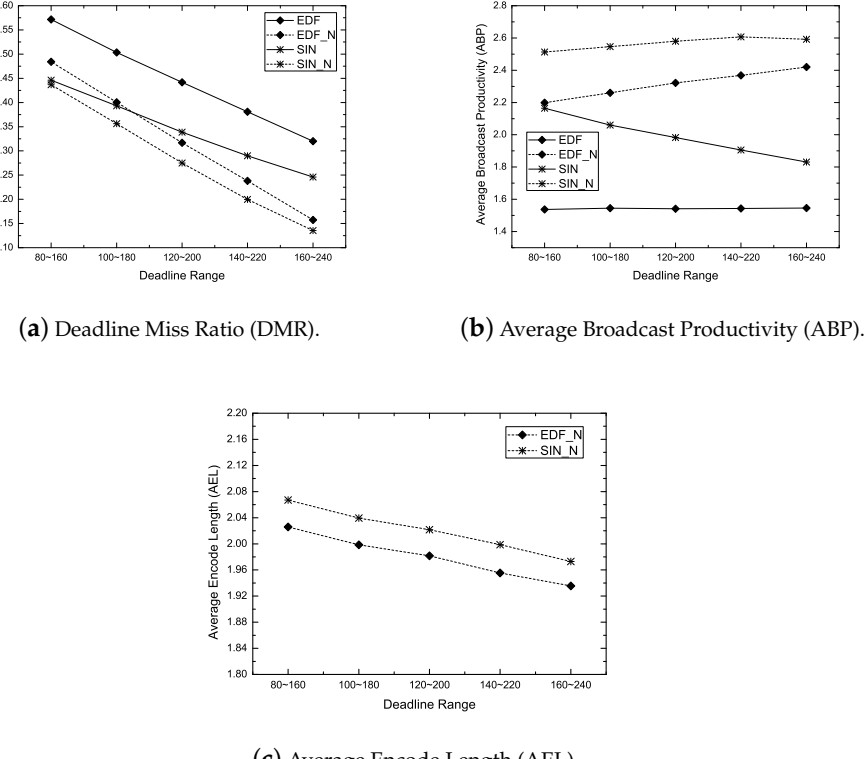

(**a**) Deadline Miss Ratio (DMR).

(**b**) Average Broadcast Productivity (ABP).

(**c**) Average Encode Length (AEL).

**Figure 3.** Impact of deadline range on the real-time model.

Figure 2b shows that with an increase number of clients, the ABP of both the coding and non-coding version increases. With the growing number of requests, the popularity of data items increases. Therefore, for the non-coding version, despite a heavier system workload, the ABP increases. For the coding version, more requests provide more coding opportunities (Figure 2c), thereby yielding a higher ABP (Figure 2b). Please note that the amount of performance improvement of using network coding for EDF is larger than SIN and the amount of performance improvement of SIN_N over SIN diminishes when the number of clients is large. This is because network coding indirectly endows the item popularity attribute to EDF when it chooses the largest clique that contains the data item with the earliest deadline. Therefore, the performance improvement of EDF_N is more significant, and it increases with the item popularity. However, since SIN has already considered the item popularity, the performance gain due to network coding in SIN_N is less significant. On the whole, SIN_N achieves the highest ABP and the lowest DMR (Figure 2a).

Figure 2c shows that SIN_N consistently achieves a higher average encode length (AEL) than EDF_N. The result shown in Figure 2d explores the reason. In particular, it shows the percentage distribution of number of items in an encoded packet under the default setting. Most of the encoded packets consist of two items, while the rest packets consist of one or three items. Specifically, SIN_N gives a higher chance of encoding more than one data items in a packet than EDF_N. For instance, 77.2% of the packets encoded by SIN_N consist of two data items, while 74.8% of the packets encoded by EDF_N consist of two data items. On the other hand, the number of items which are not encoded in SIN_N is less than that of EDF_N. Specifically, 11% of the items are not encoded in SIN_N, while 14.2% of the items are not encoded in EDF_N. In general, a higher ABP and AEL can be achieved if there are more encoded packets which consists of more data items. This has been demonstrated in Figure 2b,c, respectively.

### 6.1.2. Impact of Deadline Range

Recalling that the relative deadline of requests is uniformly selected from the range $\mu^- \sim \mu^+$. A larger value of the relative deadline gives a looser request deadline. Consequently, the DMRs of all the algorithms (Figure 3a) drop with an increase mean value of $\mu^- \sim \mu^+$. Evidently, SIN_N outperforms others.

In the coding versions, when the deadline becomes loose, the system has more opportunities to encode and disseminate data items to serve clients. As shown in Figure 3a,b, for the coding versions, the DMRs drop and the ABPs increase slightly. However, a loose request deadline does not have much impact on the AELs ($<0.1$) of EDF_N and SIN_N (Figure 3c). In the non-coding versions, when deadline range increases, the ABP of EDF does not change, whereas SIN shows a downward trend. This is because EDF only considers the request urgency, but SIN considers both the request urgency and the item popularity. As data items with less popularity may have better chance to be disseminated when the request deadline is getting looser, the ABP of SIN drops.

### 6.2. Simulation Results in Non-Real-Time Models

Figures 4 and 5 show the comparative performance graphs of the non-real-time algorithms for both the non-coding and the coding versions.

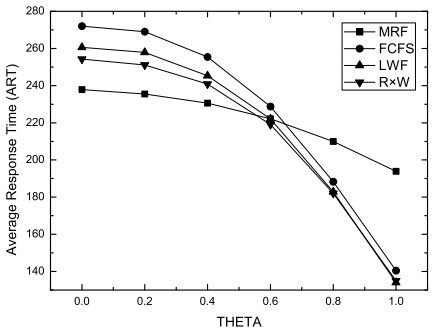

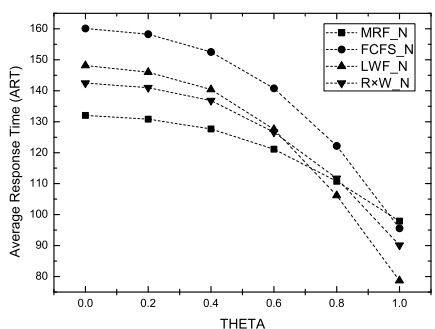

(**a**) Average Response Time (ART) of the non-coding version. (**b**) Average Response Time (ART) of the coding version.

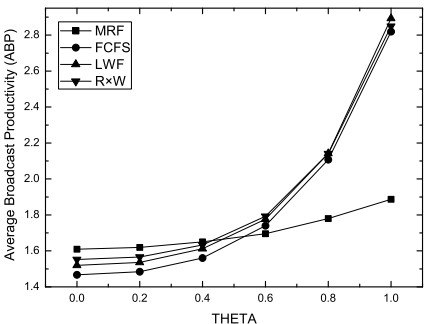

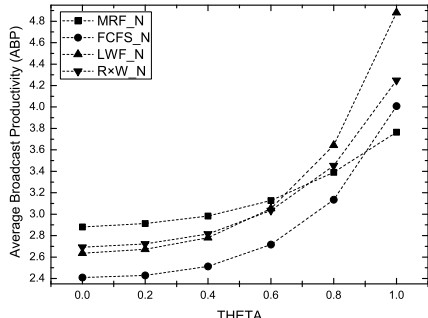

(**c**) Average Broadcast Productivity (ABP) of the non-coding version. (**d**) Average Broadcast Productivity (ABP) of the coding version.

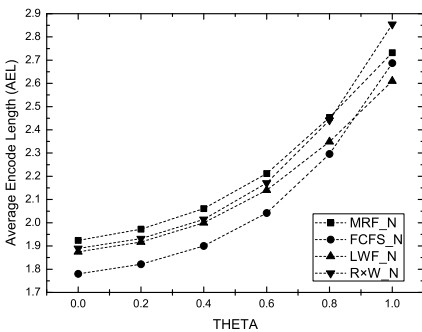

(**e**) Average Encode Length (AEL) of the coding version.

**Figure 4.** Impact of skewness parameter ($\theta$) on the non-real-time model.

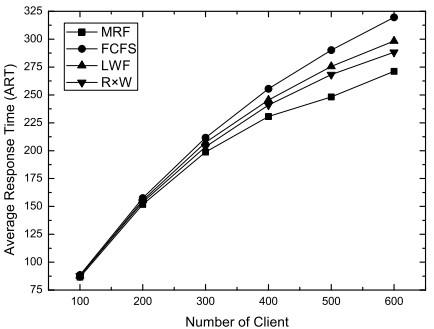
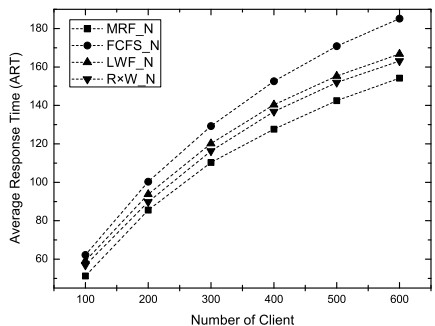

(**a**) Average Response Time (ART) of the non-coding version. 　(**b**) Average Response Time (ART) of the coding version.

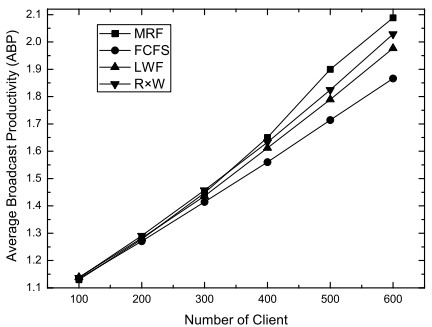
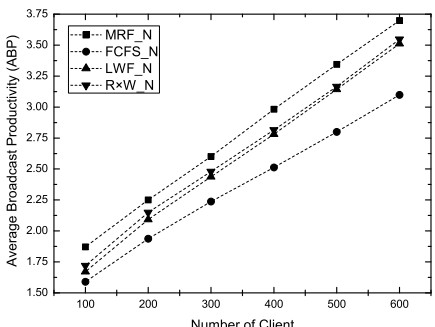

(**c**) Average Broadcast Productivity (ABP) of the non-coding version. 　(**d**) Average Broadcast Productivity (ABP) of the coding version.

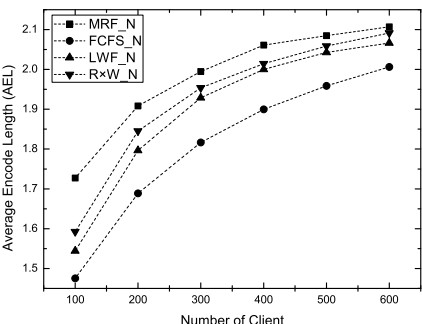

(**e**) Average Encode Length (AEL) of the coding version.

**Figure 5.** Impact of workload on the non-real-time model.

6.2.1. Impact of Skewness Parameter ($\theta$)

Figure 4 shows the impact of the data access pattern $\theta$ on the non-real-time algorithms. Evidently, both the non-coding and the coding versions of the algorithms have better performance in terms of minimizing the average response time (ART) with an increased value of $\theta$. With an increased $\theta$, the data access pattern is getting skewed. Previous studies [4,9,23] have demonstrated that algorithms will perform better when the data access pattern is getting skewed. This is because each broadcast data item has the potential to satisfy more requests. In addition, the coding version of an algorithm achieves a much lower response time (Figure 4b) than its non-coding version (Figure 4a). On an average, irrespective of the skewness of the data access pattern, the coding version of each algorithm has at

least 40% less ART than the non-coding version. The relative performance positions of the algorithms in the coding group are identical to the non-coding group. MRF performs the best when the data access pattern is less skewed. When the data access pattern is skewed, MRF becomes the worst because unpopular data items may suffer from starvation, which causes the large ART. The result is consistent with the findings in previous works [10,34,51]. MRF_N shows similar trend in Figure 4b. FCFS which only considers the request waiting time, performs the worst in most cases. Similarly, FCFS_N has the worst performance among the coding versions. R × W and LWF have similar performance, because both the algorithms consider the item popularity and the request waiting time in scheduling. In the coding version, LWF_N significantly outperforms other algorithms in a highly skewed data access pattern environment. In the non-coding version, excepting MRF, the ABPs of all the algorithms increase steadily when the data access pattern is getting skewed (Figure 4c). This is because MRF solely considers item popularity in scheduling, resulting in very low ABP for unpopular data items. Similar results are observed in the coding versions of the algorithms (Figure 4d). The AELs increase when the data access pattern is getting skewed (Figure 4e). These results further explain the relative performance of algorithms as shown in Figure 4a,b.

### 6.2.2. Impact of Workload

Figure 5 shows the impact of workload on the non-real-time scheduling algorithms. As expected, when the system workload is getting higher, the ARTs of both the non-coding and the coding versions increase (Figure 5a,b). MRF and MRF_N have the best performance in the non-coding and the coding versions, respectively. Please note that $\theta = 0.4$ in the default setting, where MRF has been shown with satisfactory performance because the data access pattern is not quite skewed. FCFS and FCFS_N perform the worst in the respective groups. R × W and R × W_N have the moderate performance in the non-coding and the coding versions, respectively. LWF and LWF_N rank next to R × W and R × W_N, respectively. Please note that the coding version of an algorithm significantly reduces the access time than its non-coding version under different workloads. It demonstrates the advantages of network coding in terms of adapting to the change of system workloads. With an increase of clients, data items will be requested more frequently. This helps to improve the broadcast productivity. Therefore, the ABPs of the non-coding versions increase (Figure 5c). Moreover, this helps to improve the coding flexibility, and hence the AELs of the coding versions increase (Figure 5e). As both the reasons positively impact on the broadcast productivity of the coding versions, when the number of clients is getting higher, the ABPs of the coding versions increase much more significantly than that of the non-coding versions (Figure 5d).

### 6.3. Simulation Results in Stretch Optimal Models

Figures 6 and 7 show the comparative performance graphs of the stretch optimal scheduling algorithms for both the non-coding and the coding versions. In this set of simulations, the sizes of data items are various. Specifically, the sizes of data items are randomly generated between 1 and 30. Moreover, for comparison purposes, we maintain a constant system workload with various data sizes by changing the *THINKTIME* accordingly.

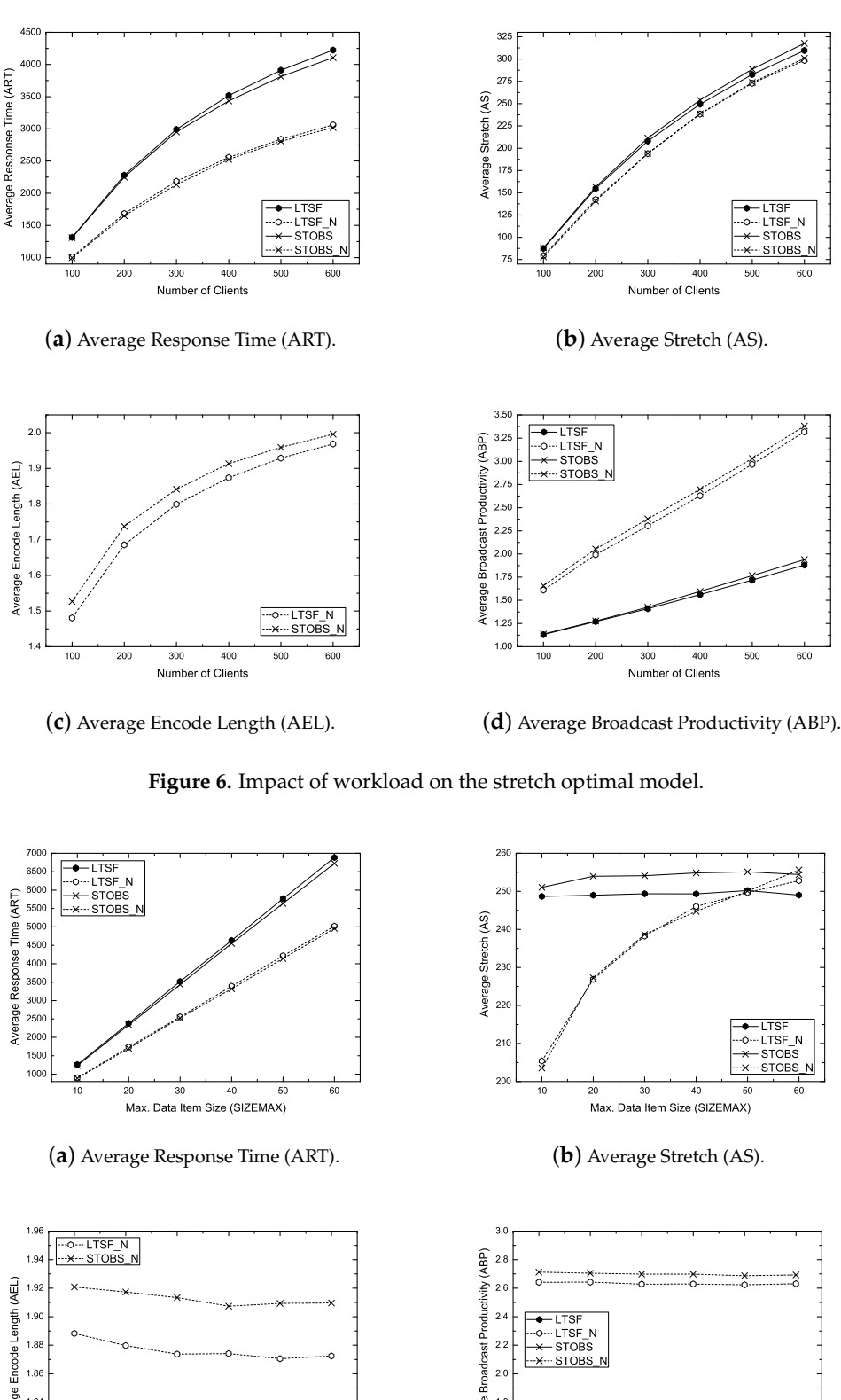

(**a**) Average Response Time (ART).  (**b**) Average Stretch (AS).

(**c**) Average Encode Length (AEL).  (**d**) Average Broadcast Productivity (ABP).

**Figure 6.** Impact of workload on the stretch optimal model.

(**a**) Average Response Time (ART).  (**b**) Average Stretch (AS).

(**c**) Average Encode Length (AEL).  (**d**) Average Broadcast Productivity (ABP).

**Figure 7.** Impact of data item size on the stretch optimal model.

### 6.3.1. Impact of Workload

Figure 6 depicts the behavior of the stretch optimal algorithms under different system workloads. The coding versions show a significant better performance than their non-coding versions in terms of minimizing the ART (Figure 6a). With respect to the average stretch (AS), a coding version also outperforms the corresponding non-coding version (Figure 6b). Both in the non-coding and the coding versions, LTSF and STOBS have similar performance. It is worth mentioning that STOBS schedules the data item with the maximum $\frac{R \times W}{S}$ value. In contrast, LTSF schedules the data item with the longest total current stretch, which is the total stretches of all the pending requests of that data item. This implies that LTSF also considers both the data item popularity and the request waiting time. Given the broadcast bandwidth of 1 unit/tick, the time to broadcast a data item (i.e., the number of broadcast ticks) equals the number of units of the data item (i.e., the item size). On a whole, the attributes of LTSF and STOBS are very similar, which result in their close performance. For the implementation of their coding versions, STOBS_N computes $\frac{N(d_s) \times t_p}{l_p}$ for finding the *selected vertex* $v_{ps}$, which is the ratio of the data item popularity multiply the maximum waiting time to the data item size. On the other hand, LTSF_N computes $\left( max \left\{ \frac{\sum t_i}{\frac{l_p}{B}} \right\} \right)$ for finding the *selected vertex* $v_{ps}$, which is the ratio of the sum of the waiting time of requests to the item size of the corresponding data item. Likewise, the attributes of STOBS_N and LTSF_N are similar, which result in their close performance. Moreover, with an increase of clients, the AEL of a coding version increases (Figure 6c), and the ABPs of both versions increase. Please note that the ABP of a coding version is much higher than its non-coding version (Figure 6d).

### 6.3.2. Impact of Data Item Size

Figure 7 shows the impact of the maximum data item size ($SIZEMAX$) on the performance of the stretch optimal algorithms. The larger value of $SIZEMAX$ means a longer service time (i.e., transmission time) of a data item. Recalling that the response time is the summation of the request waiting time and the item service time. The large $SIZEMAX$ increases the response time in two aspects. First, the transmission time is getting longer. Second, the longer service time of the current data item causes longer waiting time of other data items. As a result, with an increase of $SIZEMAX$, the ARTs of both the non-coding and the coding versions increase (Figure 7a). Since the stretch is the ratio of the response time to the transmission time of the requested data item, the stretches of the non-coding versions do not changes much because both the response time and the transmission time increase with larger $SIZEMAX$ (Figure 7b).

In contrast, with an increase value of $SIZEMAX$, the stretches of the coding versions increase. There is a tradeoff of applying network coding in scheduling for different item sizes. On the one hand, an encoded packet may consist of more than one item, which means that broadcasting an encoded packet can satisfy more pending requests simultaneously. Therefore, the average response time decreases. On the other hand, since the transmission time of an encoded packet equals the transmission time of the maximum size item of the corresponding clique, the transmission delay of an encoded packet increases, which results in the longer average response time. To sum up, whether an algorithm can take the benefit of network coding depends on which factors dominates the performance. From Figure 7a, we note that ART of a coding version is better than its non-coding version under different $SIZEMAX$ values. However, in terms of minimizing the stretch, a coding version cannot retain its superiority over the non-coding version when $SIZEMAX$ increases to 60 (Figure 7b). For instance, when $SIZEMAX$ is as low as 10, the coding versions of both LTSF and STOBS improve around 18% stretch over the non-coding versions; however, this improvement diminishes very quickly when $SIZEMAX$ increases to 40 (3% maximum improvement). Even when $SIZEMAX$ reaches to 60, the improvement becomes negative. The reason is due to the penalty of small data items, namely the coding version of an algorithm incurs higher AS for small data items. The penalty to small data items increases with the value of $SIZEMAX$. Please note that $SIZEMAX$ have no impact on the

AELs (Figure 7c). Accordingly, the ABPs of different algorithms are similar under different item sizes (Figure 7d).

## 7. Conclusions and Future Directions

On-demand data broadcast has been demonstrated as a promising way to disseminate information to a large population of clients in wireless communication environments. Conventional scheduling algorithms cannot best explore the broadcast efficiency of wireless bandwidth. In this study, we apply network coding to enhance the scheduling performance of existing on-demand scheduling algorithms. Based on the derived CR-graph, we propose a heuristic coding-based approach, which can efficiently transform existing scheduling algorithms to their respective network coding versions while preserving their original scheduling criteria. In particular, based on the classification of on-demand scheduling algorithms, we select several representative algorithms in each group for performance evaluation and enhancement, including two real-time algorithms (i.e., EDF and SIN), four non-real-time algorithms (i.e., FCFS, MRF, LWF and R×W) and two stretch optimal algorithms (i.e., LTSF and STOBS). We describe the detailed implementation of the network coding version of each algorithm and investigate their potential performance improvement using a running example. We design a number of metrics and build a simulation model to compare the performance of the non-coding version with the coding version of each algorithm. The comprehensive simulation results under various circumstances conclusively demonstrate that network coding can effectively improve the performance of on-demand scheduling algorithms with different objectives.

In our future work, we will further investigate the network coding strategy in heterogeneous service environment where data items are with different sizes. Specifically, since the benefit of network coding largely depends on the size of the encoded packet, which is determined by the largest size of data item which is encoded. It is interesting to strike a balance between encoding large and small data items to maximize the overall system performance.

**Author Contributions:** This work was completed with contributions from all the authors. V.C.S.L. and J.C. proposed the research idea with regular feedbacks of suggestions and supervision. G.G.M.N.A. helped in the conceptualization of the project, conducted further simulations and wrote the initial manuscript. Y.M. conducted the initial simulation studies. P.H.J.C. did the funding acquisition and supervised the whole project. All authors did edit, review and improve the manuscript.

**Funding:** This research received no external funding.

**Conflicts of Interest:** The authors declare no conflict of interest.

## Abbreviations

The following abbreviations are used in this manuscript:

| | |
|---|---|
| EDF | Earliest Deadline First |
| SIN | Slack time Inverse Number of pending requests |
| FCFS | Fist Come First Served |
| MRF | Most Requested First |
| LWF | Longest Wait First |
| RxW | Number of Pending Request Multiply Waiting Time |
| LTSF | Longest Total Stretch First |
| STOBS | Summary Table On-demand Broadcast Scheduler |
| CR-Graph | Cache-Request-Graph |

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
