# Peer review of "Performance Analysis of On-Demand Scheduling with and without Network Coding in Wireless Broadcast"

_futureinternet, doi:10.3390/fi11120248_

Round 1

Reviewer 1 Report

This paper proposes a heuristic coding based approach for on-demand scheduling algorithm using a graph model that consists of client node and requested data. Since network coding needs to consider the relationships between the network and the data which should be encoded together, the approach of constructing the graph using the client nodes and data seems very interesting. Intensive implementations are included in this paper, however, the reviewer has some concerns described as follows.

This paper considers the scheduling performance on different objectives and claims that the proposed approach is applicable for all the on-demand scheduling algorithms with low complexity. However, the reviewer thinks that the complexity of the decoding process is not included in the considered system. In other words, even though the proposed algorithm can deliver the network coded data with less transmissions using the scheduling algorithm, it may require more complexity to get a completely decoded source data at the client node. Therefore, if the overall complexity includes the decoding process in this paper, the proposed network coding approach may show worse performance than the "without a network coding" approach. For that reason, the decoding process of the proposed network coding system should be considered and analyzed to provide on-demand broadcast services.

One minor issue: the number of sections 4.3~4.4 need to be corrected as 4.2.2~4.2.3.

Reviewer 2 Report

This paper explores applying network coding techniques to broadcast on-demand networks. The main technique introduced is the utilization of cliques in a cache request graph. The use of cliques as part of network coding is not new, though this specific application to on-demand broadcast networks seems to be new. The formulation of the problem, proposed solution, and evaluation are sufficient to argue for the solution's merit. There are not major flaws with the work, though addressing the following comments may be warranted:

1 - A cursory search leads to numerous previous work on cache sensitive coding techniques that leverage cliques. It would be helpful to frame your work in this space.

2 - If a client receives an encode packet, it is not clear how they know which cached packet(s) they need to use to decode their desired component of the encoded packet. It seems like they would need to check up to all combinations of their cached data to find the decode key. If this is the scenario, there are some serious concerns (power set of cache is exponential). If this scenario can be avoided, discuss how the client does the decoding.

3 - There are numerous small grammatical errors throughout the paper. The only ones (in my opinion) that impede understanding are detailed below, but it would be helpful to edit the paper to make it read smoother. 

4 - Line 80. What do you mean "design the performance metrics"?

5 - Line 185 should read "...from an encoded packet if it has cached all other...".

6 - Since finding cliques is NP-Hard, it would be nice to at least roughly outline in line 305 how you find them.

7 - In line with the above comment, I assume you are not finding actual maximal cliques. If so, your running time is off. If not, you should use different wording than "find the maximum clique", since you are not actually doing that. Perhaps "find an approximately maximal clique". That could raise the critique of the algorithm you use not being an approximation algorithm (unless it is), but in my opinion, that is still better than just claiming "maximum".

Round 2

Reviewer 1 Report

The authors have addressed the comments raised by the reviewer successfully so that the manuscript can be published as a current form.